# Clinical Laboratory Utility of a Humanized Antibody in Commercially Available Enzyme Immunoassays for Coccidioidomycosis

Francisca J. Grill,[a] Collin Jugler,[a,b] Erin Kaleta,[c] Qiang Chen,[a,b] D. Mitchell Magee,[d] Thomas E. Grys,[c] Douglas F. Lake[a]

aSchool of Life Sciences, Arizona State University, Tempe, Arizona, USA

bCenter for Immunotherapy, Vaccines and Virotherapy, Biodesign Institute, Arizona State University, Tempe, Arizona, USA

cDepartment of Laboratory Medicine and Pathology, Mayo Clinic, Phoenix, Arizona, USA

dCenter for Personalized Diagnostics, Biodesign Institute, Arizona State University, Tempe, Arizona, USA

**ABSTRACT** Coccidioidomycosis, also called valley fever (VF), is a fungal infection with endemicity in desert regions of the western United States as well as certain arid regions of Central and South America. Laboratory-based diagnosis of VF often relies on the composite results from three serologic-based diagnostics, complement fixation, immunodiffusion, and enzyme immunoassay (EIA). EIA is commonly performed in clinical laboratories because results can be obtained in a few hours. Two commercially available EIAs, IMMY clarus *Coccidioides* antibody and Meridian Premier *Coccidioides*, look for the presence of anticoccidioidal IgG and IgM in patient sera that are diluted 1:441. Per regulatory requirements, this dilution step must be verified with a dilution step control despite not being provided as a reagent in either FDA-approved EIA kit. Therefore, clinical laboratories collect and reuse patient sera in subsequent tests that had a positive result in a previous test. This is a nonstandard process, reinforcing the need for a consistent and reliable dilution control. Here, we evaluate the performance of a humanized IgG and IgM antibody as a dilution control in both EIA kits. Both humanized IgG and IgM work well in each EIA and meet the appropriate threshold for positivity.

**IMPORTANCE** In southwestern and western regions of the United States, at least half a million diagnostic tests for coccidioidomycosis (valley fever) are run annually. Enzyme immunoassays (EIAs) are blood tests which require precise dilution of patient serum prior to testing. To ensure patient serum is properly diluted, there is a regulatory requirement to ensure the dilution step is accurate. Two FDA-approved EIAs used to aid in the diagnosis of coccidioidomycosis do not contain controls for this dilution step, leaving clinical laboratories with the only option of using previously positive patient sera, which may not react in a reliable or predictable manner. Here, we evaluate a humanized monoclonal antibody against a coccidioidal antigen and its utility as a dilution control in both available commercial EIAs. The use of a humanized monoclonal antibody provides a standardized and well-characterized dilution control for use in serological assays that aid in diagnosis of coccidioidomycosis.

**KEYWORDS** coccidioidomycosis, diagnostic, enzyme immunoassay, CLIA, monoclonal antibodies

Address correspondence to Douglas F. Lake, douglas.lake@asu.edu, or Thomas E. Grys, grys.thomas@mayo.edu.

The authors declare no conflict of interest.

Coccidioidomycosis, or valley fever (VF), is a fungal infection endemic in dry areas throughout the Americas caused by the inhalation of *Coccidioides* spores (1, 2). Disease manifestation varies depending on infection severity and host immune competency, with a reported 60% of patients showing no symptoms despite many having immunologic evidence of exposure (3–5). Those who are symptomatic exhibit respiratory symptoms such as cough, dyspnea, chest pain, and fatigue, all of which are difficult to distinguish from a bacterial or viral community-acquired pneumonia (6, 7). Less than

1% of patients with VF experience extrathoracic dissemination of disease to sites such as the skin, bone, and meninges (7).

Definitive diagnosis of VF is achieved through culture of *Coccidioides* or microscopic identification of the spherule form of the fungus in clinical specimens (5). However, the overall recovery rate of *Coccidioides* from over 55,000 clinical specimens sent for culture from patients with compatible symptoms was reported to be 3.2% over a 6-year period (1998 to 2003), and even in the case of confirmed coccidioidal meningitis, recovery from cerebrospinal fluid did not exceed 50% (5, 8). Thus, serological assays that detect the presence of anticoccidioidal antibodies are heavily relied upon for diagnosis of VF (7, 9). These include complement fixation, immunodiffusion (ID), and enzyme immunoassay (EIA). All three of these assays utilize some combination of *Coccidioides* antigens, including tube precipitin (TP) and complement fixation (CF) antigens, against which immunocompetent VF patients form IgM and/or IgG antibodies (10, 11). CF antigen is also called chitinase-1 (CTS1), a 47.4-kDa protein that has been characterized extensively (12–16).

The use of recombinant CTS1 (rCTS1) and truncations of rCTS1 have been tested to measure anti-CF antibodies in VF patient serum using an EIA format (15, 17, 18). Currently, there are two Food and Drug Administration (FDA)-approved EIAs used for the qualitative detection of anti-coccidioidal IgG and IgM, Premier *Coccidioides* EIA (Meridian Bioscience, Cincinnati, OH) and clarus *Coccidioides* antibody EIA (Immuno-Mycologics, Inc. [IMMY], Norman, OK). In both test kits, patient serum is diluted 1:441 for screening per manufacturer instructions. It is a Clinical Laboratory Improvement Amendments (CLIA) regulation that clinical laboratories run a dilution control alongside patient serum to ensure correct operation (19). However, dilution controls are not provided with either kit, forcing laboratories to identify and reuse previously positive serum, which is a nonstandard process. This requirement has therefore highlighted the need for consistent IgG and IgM dilution controls for clinical laboratories performing *Coccidioides* EIAs. While the composition and concentration of the antigens used in both commercial EIAs are proprietary information, previous data generated by our laboratory have demonstrated that CTS1 is a component present in complement fixation and immunodiffusion antigens supplied by IMMY and Meridian (20). This finding further suggests that CTS1 is a component of the antigenic preparation used in each EIA, whether intentional or not. Therefore, in this report, we evaluate an anti-CTS1 humanized monoclonal antibody produced in plants as chimeric IgG and IgM isotypes (21) in both IMMY and Meridian FDA-approved *Coccidioides* EIA kit tests.

## RESULTS

**Commercial enzyme immunoassay performance.** P-4H2 IgG surpassed the cutoff for positivity on the IMMY clarus *Coccidioides* antibody EIA at a concentration as low as 0.25 $\mu$g/mL, while P-4H2 IgM met the cutoff for positivity when used at a concentration greater than 1 $\mu$g/mL (Fig. 1A). P-4H2 IgG and P-4H2 IgM met the cutoff for positivity on the Meridian Premier *Coccidioides* EIA when used at a concentration greater than 0.5 $\mu$g/mL (Fig. 1B). Stocks of each antibody were prepared at concentrations such that a 1:441 dilution alongside patient samples would yield a positive result. These were validated as dilution controls by a clinical laboratory that uses the Premier *Coccidioides* EIA (Fig. 1B). The positive controls included in the IMMY and Meridian kits did not provide adequate signal at a 1:441 dilution and are not suitable to be used as dilution controls (Fig. 2).

**Western blot detection of CTS1 in commercial immunodiffusion antigens.** After performing SDS-PAGE under reducing conditions, Western blotting was performed using P-4H2 IgG and IgM. CTS1 was detected by Western blotting in *Coccidioides posadasii* strain Silveira mycelial supernatant, IMMY *Coccidioides* IDCF, IMMY *Coccidioides* IDTP, and Meridian "F" ID antigens, but was not detected in Meridian "TP" ID antigen (Fig. 3). No CTS1 was detected in *Aspergillus*, *Blastomyces*, or *Histoplasma* immunodiffusion preparations from IMMY (Fig. 3).

## DISCUSSION

The humanized plant-produced monoclonal antibodies (MAbs) P-4H2 IgG and P-4H2 IgM were shown to bind as predicted in the two commercially available EIAs that are used to aid

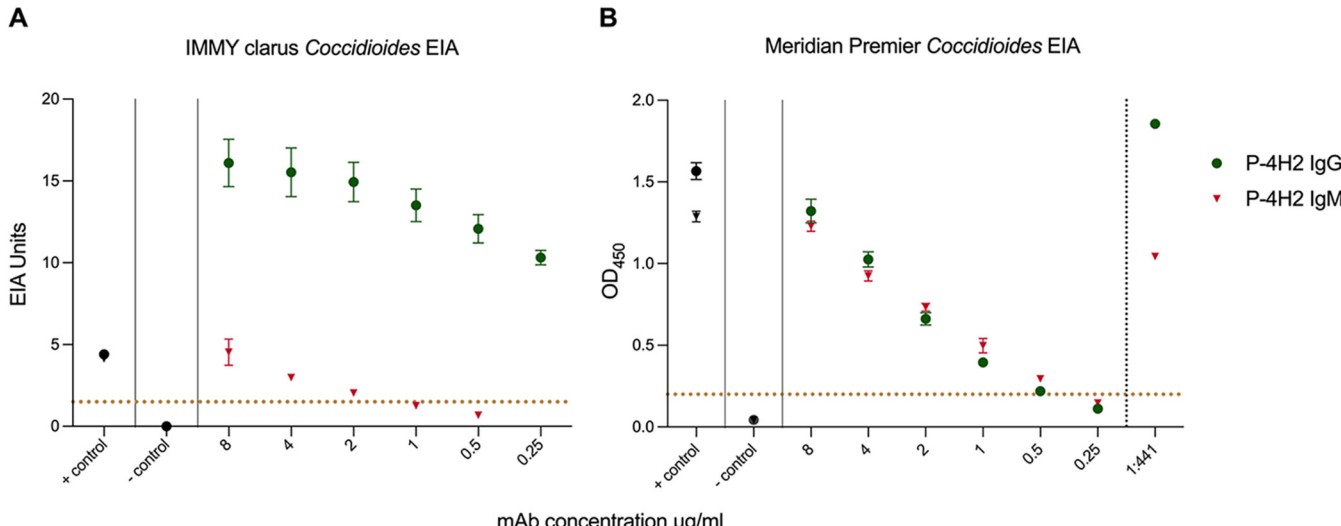

**FIG 1** Performance of P-4H2 IgG and P-4H2 IgM on IMMY clarus *Coccidioides* EIA (A) and Meridian Premier *Coccidioides* EIA (B). Results for IMMY clarus *Coccidioides* EIA are reported in calculated EIA units, with a positive cutoff at 1.5 EIA units for both IgG and IgM, shown with a dotted yellow line. Results for Meridian Premier *Coccidioides* EIA are reported as $OD_{450}$ values, with a positive cutoff value of 0.2 for both IgG and IgM, shown with a dotted yellow line. The positive and negative controls provided in the kit of each EIA are colored black. Means ± SDs is plotted from three replicates.

in the diagnosis of coccidioidomycosis. Both EIAs are FDA approved; however, neither kit includes dilution controls for IgG or IgM so that laboratories can satisfy the CLIA requirement of diluting patient samples in the same manner as known IgG and IgM controls. To meet this requirement, clinical laboratories must collect and reuse patient sera that were previously IgG and IgM positive for use as dilution controls in subsequent runs. This may prove problematic not only because the stored serum is a limited and finite source but prolonged storage may lead to decreased reactivity on EIA and, subsequently, invalid results if the positive cutoff threshold is not met. Additionally, an error such as a misdilution (e.g., 1:21 instead of 1:441) may be difficult to detect using patient serum as the dilution control since this would still produce a positive result. Inclusion of standardized dilution controls at a known concentration

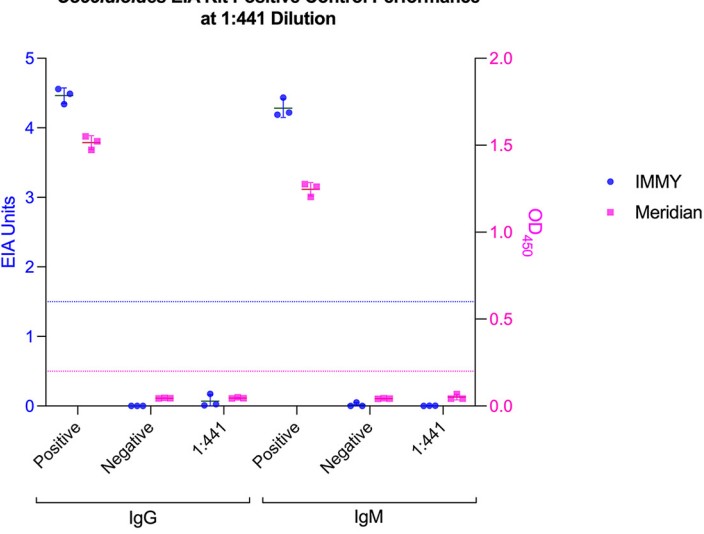

**FIG 2** Performance of *Coccidioides* EIA kit-provided positive controls at a 1:441 dilution. IMMY clarus *Coccidioides* EIA is represented with blue circles, with values reported in EIA units. Meridian Premier *Coccidioides* EIA is represented with pink squares, with values reported as $OD_{450}$ measurements. The cutoff line for positivity is represented for each in its respective color (1.5 EIA units for IMMY; $OD_{450}$ of 0.2 for Meridian). Means (horizontal bar) ± SDs (error bars) are plotted from three replicates. Nondiluted positive and negative controls were used as provided in the kits.

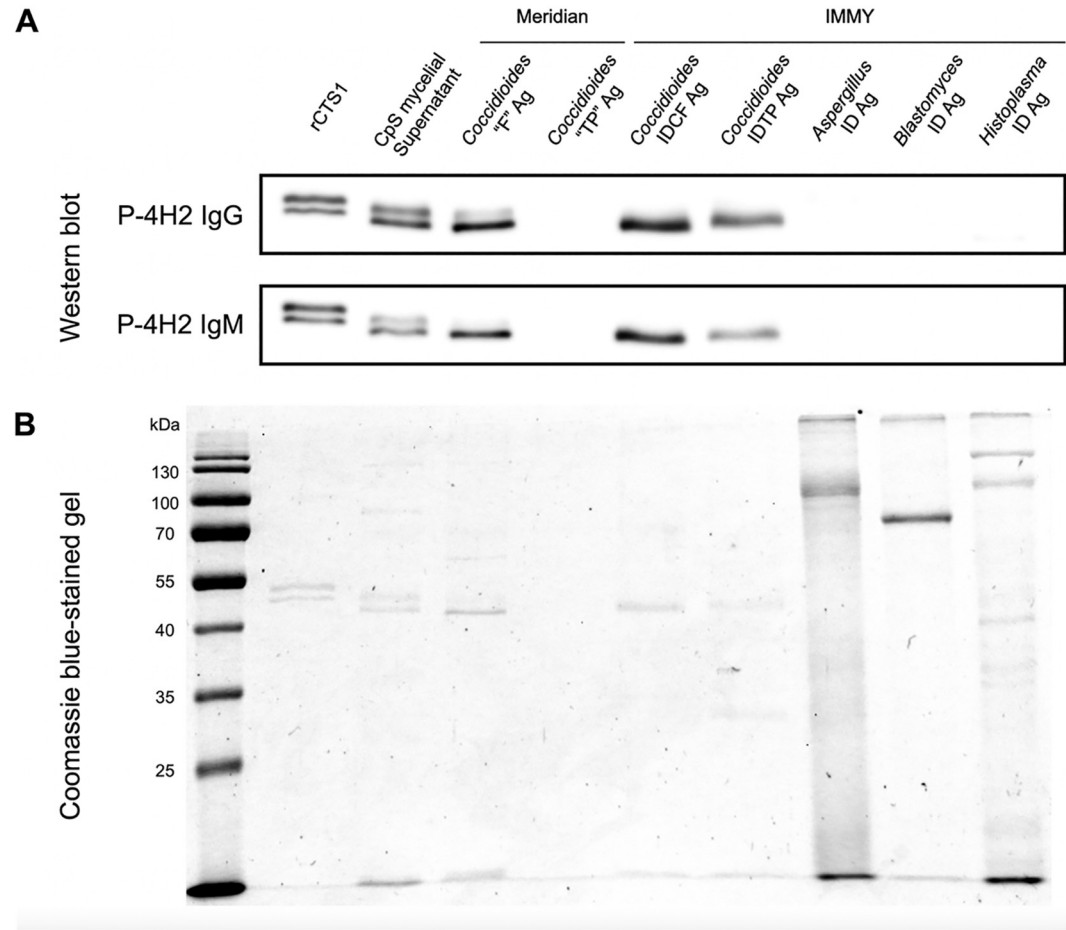

**FIG 3** (A) Detection of *Coccidioides* CTS1 in commercial immunodiffusion antigen preparations by P-4H2 IgG and P-4H2 IgM on Western blotting. (B) Coomassie blue-stained electrophoresed proteins in each antigen preparation. Images taken on an Azure 600 imager (Azure Biosystems).

would result in an expected optical density range so that if the result falls outside that range, a dilution error may have occurred, and the test should be repeated. Ultimately, reuse of previously positive samples is a nonstandard practice and is not included in manufacturer instructions.

Our results showing that recombinant plant-produced anti-CTS1 humanized MAbs can serve as dilution controls for both IMMY and Meridian EIAs suggests that CTS1 is a major component of the antigen preparations used in the EIA kits. The Meridian Premier *Coccidioides* EIA provides microwells coated with antigens that are used for both IgG and IgM detection, which are differentiated with the enzyme-conjugated secondary antibody used, either anti-human IgG or anti-human IgM. In contrast, the IMMY clarus *Coccidioides* EIA provides separate microwells coated with different antigens (termed "CF antigen-coated" and "TP antigen-coated" microwells) in addition to anti-human IgG and IgM enzyme-conjugated secondary antibodies for use only on their respective wells. The antigenic contents of the CF- and TP-antigen coated wells are proprietary; however, the lower reactivity of P-4H2 IgM on the IMMY TP antigen-coated wells suggests that there is less CTS1 present on these wells, as P-4H2 IgG and IgM bind specifically to CTS1/CF antigen.

Interestingly, CTS1 appears to be a component in both the *Coccidioides* CFID and TPID antigens distributed by IMMY, while CTS1 was not detectable in Meridian's *Coccidioides* "TP" immunodiffusion (ID) antigen (Fig. 3). A previous study that retrospectively compared complement fixation, ID, and EIA results in VF patients found that patients had various levels of seropositivity in each assay, but when all methods were considered together, the rate of seropositivity was higher than for any individual assay (22). This lack of a clear reference

standard is why evaluations of *Coccidioides* serological methods must employ a composite clinical reference standard (23). For vendors, lot-to-lot composition and performance could be better adjusted for consistent products by employing the recombinant MAbs P-4H2 IgG and IgM reported here. Similarly, clinical laboratories could use the MAbs to ensure consistent performance of their assays and to help troubleshoot unexpected findings and determine likely sources of variability (reagent, instrument, operator, etc.).

Overall, our results demonstrate the utility of new reagents that have the potential to meet the need for assay dilution controls as well as provide semiquantitative insights into the ongoing assay performance of EIA kits to aid in the serological diagnosis of coccidioidomycosis. We envision providing each antibody as a reagent for use in both *Coccidioides* EIA kits to run alongside patient samples to confirm proper dilution and operation of these assays.

## MATERIALS AND METHODS

**Generation of humanized antibodies.** The variable regions of 4H2, a murine monoclonal antibody (MAb) to CTS1, were cloned, sequenced, and genetically fused into human IgG and IgM Fc-containing plasmids to form a chimeric mouse V-region–human Fc region antibody construct, previously detailed (21). The chimeric plasmids were subjected to agroinfiltration in *Nicotiana benthamiana* leaves, followed by harvest and purification as described elsewhere (21, 24). The resultant plant-produced humanized MAbs (P-4H2 IgG and P-4H2 IgM) were evaluated for their performance on two commercially available enzyme immunoassays (EIAs) that detect human IgG and IgM antibodies specific for *Coccidioides* spp., described below. The humanized antibodies were also evaluated for the ability to detect CTS1 in immunodiffusion commercial antigen preparations.

**Commercial enzyme immunoassays for coccidioidomycosis.** P-4H2 IgG and P-4H2 IgM were tested in Meridian Bioscience Premier *Coccidioides* EIA and IMMY clarus *Coccidioides* antibody EIA according to the instructions and reagents provided in each kit. Meridian's EIA kit contains the same microtiter wells for both IgG and IgM screening, while IMMY's EIA kit contains two types of microtiter wells, CF antigen-coated wells for IgG screening and TP antigen-coated wells for IgM screening. Briefly, 2-fold dilutions ranging from 8 $\mu$g/mL to 0.25 $\mu$g/mL of P-4H2 IgG and IgM were prepared in 1× specimen diluent and incubated in microtiter wells for 30 min alongside positive and negative controls provided with each kit. IMMY kits also contain a calibrator cutoff solution that is used to establish the cutoff signal used to calculate EIA units. Plates were washed three times with freshly made 1× wash buffer followed by addition of corresponding anti-IgG or anti-IgM conjugated antibody, also provided with each kit. After 30 min, plates were washed three times, and the substrate was added. The Meridian EIA was allowed to develop for 5 min before addition of stop solution, while the IMMY EIA developed for 10 min before addition of stop solution. Results of the Meridian EIA are reported as optical density at 450 nm ($OD_{450}$) values, while IMMY results are reported in EIA units calculated using the provided CF or TP calibrator cutoff.

**Western blotting of immunodiffusion antigens.** The amount of CTS1 in commercial antigen preparations used for immunodiffusion was quantified using our previously published assay (20) in order to normalize the amount of CTS1 between antigens. Based on the amount quantified in each antigen, 40 ng of CTS1 was prepared with 4× sodium dodecyl-sulfate (SDS) sample loading buffer for subsequent SDS-polyacrylamide gel electrophoresis (PAGE). For antigens that did not have any measurable amount of CTS1, total protein concentration was quantified with Pierce bicinchoninic acid (BCA) protein assay kit (Thermo Scientific) according to the manufacturer's instructions. If the measurable amount of protein was more than 250 $\mu$g/mL, 5 $\mu$g total protein was prepared; if less than 250 $\mu$g/mL, 10 $\mu$L was prepared. Electrophoresed proteins were either stained with Coomassie blue dye or transferred onto polyvinylidene difluoride (PVDF) membranes (Thermo Scientific) using a Western blotting apparatus (Bio-Rad). Membranes were blocked in 1% bovine serum albumin (BSA) in Tris-buffered saline with 0.1% Tween 20 (TBST) overnight at 4°C. Anti-CTS1 MAbs P-4H2 IgG and P-4H2 IgM were diluted to 0.5 $\mu$g/mL in 1% BSA-TBST and incubated on membranes for 1 h. Membranes were washed three times in TBST followed by addition of the appropriate secondary antibody. For P-4H2 IgG, goat anti-human IgG Fc-specific horseradish peroxidase (HRP; Sigma-Aldrich) was used at 1:50,000; for P-4H2 IgM, goat anti-human IgM Mu-HRP (Caltag) was used at 1:50,000. Secondary antibodies were allowed to incubate for 45 min followed by four washes in TBST. Membranes were washed two additional times with TBS before development with SuperSignal West Dura extended-duration chemiluminescent substrate (Thermo Scientific). Images were taken on an Azure600 imager (Azure Biosystems).

## ACKNOWLEDGMENTS

We thank Stephanie Boardman and the members of the core lab at Mayo Clinic Arizona for testing our antibody controls on the Meridian Premier *Coccidioides* EIA.

D.F.L. and T.E.G. conceptualized research; F.J.G. and C.J. designed and performed experiments. F.J.G. analyzed the data and wrote the paper with revisions from Q.C., E.K., D.M.M., T.E.G., and D.F.L.

F.J.G., T.E.G., and D.F.L. are inventors on an unlicensed patent application involving the MAbs evaluated in this report. All other authors declare no conflicts of interest.

Funding to support this work was provided by grants from NIAID 1R21AI152042 to D.F.L. and T.E.G. and 1R01AI155954 to D.F.L. and D.M.M.

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
