## [Reviewer comments · Microbiology Spectrum]

Microbiology Spectrum

Clinical laboratory utility of a humanized antibody in commercially available enzyme immunoassays for coccidioidomycosis

Francisca Grill, Collin Jugler, Erin Kaleta, Qiang Chen, D. Mitch Magee, Thomas Grys, and Douglas Lake

Corresponding Author(s): Douglas Lake, Arizona State University

Review Timeline:

Submission Date:	July 6, 2022
Editorial Decision:	August 7, 2022
Revision Received:	August 16, 2022
Accepted:	September 4, 2022

Editor: Agostinho Carvalho

Reviewer(s): Disclosure of reviewer identity is with reference to reviewer comments included in decision letter(s). The following individuals involved in review of your submission have agreed to reveal their identity: Joshua Fierer (Reviewer #2)

Transaction Report:

DOI: <https://doi.org/10.1128/spectrum.02573-22>

August 7, 2022

Dr. Douglas F. Lake
Arizona State University
School of Life Sciences
13208 E. Shea Blvd MCCR2 2-228
Mayo Collaborative Research Building
Scottsdale, AZ 85259

Re: Spectrum02573-22 (Clinical laboratory utility of a humanized antibody in commercially available enzyme immunoassays for coccidioidomycosis)

Dear Dr. Douglas F. Lake:

Link Not Available

Sincerely,

Agostinho Carvalho

Journals Department
Reviewer comments:

Reviewer #1 (Comments for the Author):

Here, the authors investigate the utility of humanized IgG and IgM antibodies against the CTS1 protein of *Coccidioides* in serologic assays for Valley Fever. Using two commercial serology enzyme immunoassay (EIA) kits, the authors show that their IgG and IgM are detected by the kits in a dose-dependent fashion. Such an observation supports the use of these reagents as reference standards in these assays. The authors further demonstrate engagement of immunodiffusion antigen reagents with their antibodies using Western blotting. Overall, the authors conclude that their antibodies represent a better method for standardizing Valley Fever enzyme immunoassays relative to the current practice of re-using seropositive patient serum.

Specific Comments:

1. Given the broad audience of Microbiology Spectrum, the authors need to explain the differing enzyme immunoassay formats more thoroughly and how those differences influence their results. For example, the IMMY kit contains two antigen reagents, CF and TP. Can these data be presented separately in the figures? Why would only the IgM show lower reactivity to the TP reagent when both the IgG and IgM target the same antigen? Does this reduced IgM reactivity underly the comparatively lower signal for IgM in the IMMY assay? The assay setups should be more clearly described.
2. How do these new antibody reagents differ from the positive controls used in each assay? Why can these positive controls not be used as dilution controls? This should be elaborated. To this end, the authors should include a brief statement in the discussion describing exactly what type of reagent they envision for use as a dilution standard. Would this be a common serum spiked with known concentrations of antibody or simply a solution of the antibody in buffer that is then diluted?
3. It would be beneficial to include a representative dilution curve of serum from a seropositive patient in each assay to see how the antibodies compare to the current standard practice.

Reviewer #2 (Comments for the Author):

This paper by Grill and colleagues reports the development and potential use of a humanized mouse monoclonal antibody against chitinase 1 from *Coccidioides* sp. as a control reagent that can be used with 2 of the best serological commercial EIA assays to diagnose coccidioidomycosis. Both assays require a complex dilution scheme. Best practice for that step is to run a positive control system in parallel but the companies do not provide a positive control. Thus, either the control is omitted, or each laboratory needs to use a non-standardized control. Since the antigens used in the commercial assays are proprietary secrets, they first used their monoclonals to demonstrate that the CF antigen (chitinase 1) was present in the IgG and IgM assays of both manufacturers' kits. They then showed that their monoclonals could detect the chitinase 1 in all four assays at the required dilution, and therefore could be used as a positive control.

The authors never explain why dilution is necessary. Since they have previously used their monoclonal to detect circulating chitinase in serum, could it be to dilute out the competing antigen?

This work is of interest to laboratories and presumably to the manufactures of the commercial tests. Since the antibodies to chitinase 1 made by people are apparently to a conformational epitope as shown by Galgiani's group (doi: 10.1016/j.diagmicrobio.2020.115198), it is possible that the mouse monoclonal may recognize only one possible immune response to that protein, there may be a discordance between a patient's result and the monoclonal control, in either direction.

Reviewer #3 (Comments for the Author):

Authors evaluated two recombinant humanized monoclonal antibodies against a coccidioidal antigen in two commercial immunoassays detecting anti-coccidioidal antibodies. The premise of this study is that the two commercial kits require a 1:441 dilution of serum (through two serial 1:20 dilutions) but the kits don't provide reagents to control for the dilution. The author's aim was to show that their antibody reagents could be used as dilution control. However, they only show that their reagents result in a positive result. They do not provide evidence showing their reagent is able to detect a miss dilution such as 1:20.

Line 12 Do you mean performance of a humanized IgG and IgM antibody as dilution control? It is unclear. The importance statement more clearly describes the aim of the study.

Line 29 define VF

Line 48 What about the Sona Lateral Flow Assay by IMMY? It also requires a 1:441 dilution

Any reason why the positive control provided in the kit can't be diluted and used a dilution control?

Staff Comments:

Preparing Revision Guidelines

To submit your modified manuscript, log onto the eJP submission site at <https://spectrum.msubmit.net/cgi-bin/main.plex>. Go to

Author Tasks and click the appropriate manuscript title to begin the revision process. The information that you entered when you first submitted the paper will be displayed. Please update the information as necessary. Here are a few examples of required updates that authors must address:

Please return the manuscript within 60 days; if you cannot complete the modification within this time period, please contact me. If you do not wish to modify the manuscript and prefer to submit it to another journal, please notify me of your decision immediately so that the manuscript may be formally withdrawn from consideration by Microbiology Spectrum.

Response to reviewers' comments:

We would like to thank the reviewers for their comments on our manuscript "Clinical laboratory utility of a humanized antibody in commercially available enzyme immunoassays for coccidioidomycosis". We have discussed and carefully reviewed these comments and suggestions and have revised the manuscript accordingly. Changes in the manuscript are highlighted with Tracked Changes and responses to individual comments can be found below. Please note that references to lines are for the Tracked Changes copy of the manuscript.

1 Reviewer #1

1.1 Comment: Given the broad audience of Microbiology Spectrum, the authors need to explain the differing enzyme immunoassay formats more thoroughly and how those differences influence their results. For example, the IMMY kit contains two antigen reagents, CF and TP. Can these data be presented separately in the figures? Why would only the IgM show lower reactivity to the TP reagent when both the IgG and IgM target the same antigen? Does this reduced IgM reactivity underly the comparatively lower signal for IgM in the IMMY assay? The assay setups should be more clearly described.

Response: To clarify, the CF and TP antigens used by IMMY are already coated on the wells, not provided as reagents. The contents of each are proprietary, however previous literature has suggested CF antigen is CTS1 (lines 45-46), while TP antigen is less well defined. We have added an explanation of the different antigens coated on the microtiter wells (lines 80-82).

The lower reactivity of our monoclonal antibody on IMMY's TP antigen-coated wells suggests that perhaps only a small amount of CTS1 is present on the TP antigen-coated wells (lines 146-151). Our humanized IgG and IgM target the same antigen (CTS1) but patient IgG and IgM may not.

We have made a figure for reviewer's eyes only to elucidate the setup. We feel that this figure does not need to be included in the manuscript as we have written it out clearly in the methods (lines 78-91).

1.2 Comment: How do these new antibody reagents differ from the positive controls used in each assay? Why can these positive controls not be used as dilution controls? This should be elaborated. To this end, the authors should include a brief statement in the discussion describing exactly what type of reagent they envision for use as a dilution standard. Would this be a common serum spiked with known concentrations of antibody or simply a solution of the antibody in buffer that is then diluted?

Response: The positive control provided by IMMY is "Anti-*Coccidioides* antibodies." The positive control provided by Meridian is "Prediluted positive human serum" (information obtained from package inserts). We tested these positive controls at a 1:441 dilution and they do not produce a positive signal and therefore cannot be used as dilution controls (lines 122-123). We've added a figure (Figure 2) to reflect these findings. Our antibodies differ because they are recombinantly produced, characterized antibodies (anti-CTS1 monoclonal chimeric IgG and IgM antibodies), at known concentrations.

We envision providing each antibody at a known concentration in a protein-buffered solution that is then diluted in the dilution buffers provided in each kit, just as patient serum is. We have added a brief statement in the discussion stating this (lines 165-167).

1.3 Comment: It would be beneficial to include a representative dilution curve of serum from a seropositive patient in each assay to see how the antibodies compare to the current standard practice.

Response: We disagree that this would be beneficial, as we know the concentration of our monoclonal antibody while we do not know the concentration of anti-CTS1 antibodies present in patient serum, diminishing the standardization aspect we are striving to accomplish.

2 Reviewer #2

2.1 Comment: The authors never explain why dilution is necessary. Since they have previously used their monoclonal to detect circulating chitinase in serum, could it be to dilute out the competing antigen?

Response: Dilution of patient serum is necessary per manufacturer instructions (lines 51-52). We have not used this monoclonal antibody to detect circulating chitinase in serum.

2.2 Comment: This work is of interest to laboratories and presumably to the manufactures of the commercial tests. Since the antibodies to chitinase 1 made by people are apparently to a conformational epitope as shown by Galgiani's group (doi: 10.1016/j.diagmicrobio.2020.115198), it is possible that the mouse monoclonal may recognize only one possible immune response to that protein, there may be a discordance between a patient's result and the monoclonal control, in either direction.

Response: We've found that this monoclonal antibody does not compete for the conformational epitope that patient antibodies bind to (data not shown). The monoclonal control is to ensure proper operation of the assay, not to determine a patient's result.

3 Reviewer #3

3.1 Comment: They do not provide evidence showing their reagent is able to detect a miss dilution such as 1:20.

Response: This is an excellent point. Regardless of what reagent is used as a dilution control (our monoclonal antibody or patient serum), this could be very hard for a clinical laboratory to detect.

Because we know the expected OD value range for our monoclonal, a mis-dilution would result in an abnormally high OD. We could provide the control at a concentration where we know the expected result range. We have highlighted this additional benefit in our discussion (lines 141-142 & 154-155).

3.2 Comment: Line 12 Do you mean performance of a humanized IgG and IgM antibody as dilution control? It is unclear. The importance statement more clearly describes the aim of the study.

Response: Thank you for bringing this to our attention. You are correct and this is reflected in the change on line 13.

3.3 Comment: Line 29 define VF

Response: VF has now been defined on line 29.

3.4 Comment: Line 48 What about the Sona Lateral Flow Assay by IMMY? It also requires a 1:441 dilution

Response: Our paper is focusing on the *Coccidioides* enzyme immunoassays. While it would be beneficial to try the performance of our controls on the Sona Lateral Flow Assay, this is less widely used and is beyond the scope of our paper.

3.5 Comment: Any reason why the positive control provided in the kit can't be diluted and used a dilution control?

Response: The positive control provided in each kit does not work when diluted 1:441. Please refer to response above to Reviewer #1 (Comment 1.2)

September 4, 2022

Dr. Douglas F. Lake
Arizona State University
School of Life Sciences
13208 E. Shea Blvd MCCR2 2-228
Mayo Collaborative Research Building
Scottsdale, AZ 85259

Re: Spectrum02573-22R1 (Clinical laboratory utility of a humanized antibody in commercially available enzyme immunoassays for coccidioidomycosis)

Dear Dr. Douglas F. Lake:

Your revised manuscript has been evaluated by the external referees. They agree that the manuscript is now suitable for publication in *Microbiology Spectrum*.

Your manuscript has been accepted, and I am forwarding it to the ASM Journals Department for publication. You will be notified when your proofs are ready to be viewed.

As an open-access publication, *Spectrum* receives no financial support from paid subscriptions and depends on authors' prompt payment of publication fees as soon as their articles are accepted. You will be contacted separately about payment when the proofs are issued; please follow the instructions in that e-mail. Arrangements for payment must be made before your article is published. For a complete list of **Publication Fees**, including supplemental material costs, please visit our website.

Thank you for submitting your paper to *Spectrum*.

Sincerely,

Agostinho Carvalho
Editor, *Microbiology Spectrum*
